# Small Non-Coding RNAome of Ageing Chondrocytes

**DOI:** 10.3390/ijms21165675

**Published:** 2020-08-07

**Authors:** Panagiotis Balaskas, Jonathan A. Green, Tariq M. Haqqi, Philip Dyer, Yalda A. Kharaz, Yongxiang Fang, Xuan Liu, Tim J.M. Welting, Mandy J. Peffers

**Affiliations:** 1Institute of Life Course and Medical Sciences, William Henry Duncan Building, 6 West Derby Street, Liverpool L7 8TX, UK; pddyer@liv.ac.uk (P.D.); yalda@liverpool.ac.uk (Y.A.K.); 2Department of Anatomy and Neurobiology, Northeast Ohio Medical University, Rootstown, OH 44272, USA; dun06hpu@googlemail.com (J.A.G.); thaqqi@neomed.edu (T.M.H.); 3Centre for Genomic Research, Institute of Integrative Biology, Biosciences Building, Crown Street, University of Liverpool, Liverpool L69 7ZB, UK; fangy@liv.ac.uk (Y.F.); xuanliu@liverpool.ac.uk (X.L.); 4Department of Orthopaedic Surgery, Maastricht University Medical Centre, 6202 AZ Maastricht, The Netherlands; t.welting@maastrichtuniversity.nl

**Keywords:** chondrocyte, ageing, equine, small non-coding RNA

## Abstract

Ageing is a leading risk factor predisposing cartilage to osteoarthritis. However, little research has been conducted on the effect of ageing on the expression of small non-coding RNAs (sncRNAs). RNA from young and old chondrocytes from macroscopically normal equine metacarpophalangeal joints was extracted and subjected to small RNA sequencing (RNA-seq). Differential expression analysis was performed in R using package DESeq2. For transfer RNA (tRNA) fragment analysis, tRNA reads were aligned to horse tRNA sequences using Bowtie2 version 2.2.5. Selected microRNA (miRNAs or miRs) and small nucleolar RNA (snoRNA) findings were validated using real-time quantitative Polymerase Chain Reaction (qRT-PCR) in an extended cohort of equine chondrocytes. tRNA fragments were further investigated in low- and high-grade OA human cartilage tissue. In total, 83 sncRNAs were differentially expressed between young and old equine chondrocytes, including miRNAs, snoRNAs, small nuclear RNAs (snRNAs), and tRNAs. qRT-PCR analysis confirmed findings. tRNA fragment analysis revealed that tRNA halves (tiRNAs), tiRNA-5035-GluCTC and tiRNA-5031-GluCTC-1 were reduced in both high grade OA human cartilage and old equine chondrocytes. For the first time, we have measured the effect of ageing on the expression of sncRNAs in equine chondrocytes. Changes were detected in a number of different sncRNA species. This study supports a role for sncRNAs in ageing cartilage and their potential involvement in age-related cartilage diseases.

## 1. Introduction

Articular cartilage is a specialised connective tissue of diarthrodial joints. Its smooth lubricated surface assists joint movement and its mechanical properties facilitate load bearing in the joint. The tissue harbours one cell type; the chondrocyte, and is devoid of blood vessels and nerves, receiving nutrients from synovial fluid and subchondral bone [1]. Articular cartilage is characterised by an extracellular matrix (ECM) consisting of mainly collagen type 2 and proteoglycans, which give the tissue many of its properties. After reaching maturity, cartilage displays a limited repairing capacity as indicated by low chondrocyte proliferation and low collagen turnover [2].

There are a number of factors affecting the homeostatic properties of cartilage such as genetics and obesity [3]. However, ageing is a leading risk factor that predisposes cartilage to pathological changes and disease, such as osteoarthritis (OA), the most common joint disease [3]. These age-related changes affect both chondrocyte physiology and ECM properties. Aged chondrocytes display increased senescence and higher expression of catabolic markers; features also evident in OA chondrocytes [3,4]. Moreover, in humans, aged knee cartilage is thinner compared to younger cartilage and is characterised by increased collagen crosslinking and altered proteoglycan content. These changes affect matrix stiffness, make cartilage susceptible to fractures and lower its ability to sense mechanical stimuli [3,5,6].

The exact mechanisms through which age can affect cartilage health remain elusive, though it is believed to be a cumulative combination of many molecular pathways rather than a single aetiology. Recent advances in the field have recognised epigenetics in ageing and diseased articular cartilage as an area of growing interest [6,7]. A class of epigenetic modifications that have attracted increasing attention are small non-coding RNAs (sncRNAs). They are short, typically <200 bp, RNA species, which are not translated into protein but have other structural or regulatory biological roles. These include microRNAs (miRNAs or miRs), small nuclear RNAs (snRNAs), small nucleolar RNAs (snoRNAs), piwi-interacting RNAs and transfer RNAs (tRNAs). SncRNAs are promising candidates for targeted therapeutics due to their small size and diverse cellular functions. By using specific synthetic oligonucleotides, aberrant expression of sncRNAs in disease could be modified, resulting in delay or reversal of pathological changes [8]. Furthermore, sncRNAs could be used as biomarkers to monitor disease initiation and progression or response to treatment [9,10]. This is of high importance in OA, as treatment is currently symptomatic and most patients with end-stage OA require joint replacement; a procedure with a high social and economic burden for patients and the healthcare system respectively [11]. 

MiRNAs, the most studied sncRNAs, regulate gene expression by binding complementary sequences in the 3’ untranslated region of their messenger RNA (mRNA) targets, thus inhibiting mRNA translation [8]. MiRNAs have been linked to ageing and diseased cartilage; miR-140 is important for cartilage development and deletion of miR-140 in mice causes skeletal defects [12]. Moreover, OA chondrocytes show decreased expression of miR-24, resulting in increased expression of the senescent marker p16INK4a, highlighting the link between OA and senescence; a hallmark of ageing [4].

In addition to miRNAs, snoRNAs are increasingly studied in ageing and OA. Snora73 expression increases in the joint and serum of old mice compared to young *mice* [9] and we have also previously identified a catalogue of age-related snoRNAs in human knee cartilage [13]. SnoRNAs have canonical roles in the post transcriptional modification of RNA substrates including ribosomal RNAs, and mRNAs, but can also exhibit non-canonical functions such as miRNA-like activity [14]. Their aberrant expression has also been associated with the development of some diseases [15]. Our previously conducted mouse study demonstrated alterations in the snoRNA profile of young, old and OA joints in mice when compared to healthy controls, highlighting the potential of snoRNAs to be used as novel markers for this disease [9]. We have also identified changing snoRNA profiles in ageing and OA human cartilage [16], synovial fluid from horses with early OA and diseased anterior cruciate ligaments from OA joints [17].

tRNAs are adaptor molecules of ~73–90 nucleotides long consisting of a T-loop, D-loop, variable loop, and the anticodon loop. Protein translation requires amino acids to be linked together into polypeptides and tRNAs recruit these amino acids to the translating ribosome. Recent studies have shown that tRNAs are a major source of sncRNAs with an active role in gene regulation [18]. tRNA fragments result from specific processing of tRNAs. These include tRNA halves (tiRNAs), which are 28–36 nucleotide long fragments formed by Angiogenin (ANG). ANG divides the tRNA into two halves at the anticodon loop [19] giving rise to the 3′ tiRNA and 5′ tiRNA halves. tRNAs are also processed into smaller fragments; tRNA-derived small RNA fragments (tRFs); tRF-1, tRF-2, tRF-3 and tRF-5, however the naming conventions of these classes is still not consistent [20]. RNase Z, or its cytoplasmic homologue ELAC2, [21] cleaves the 3′ trailer fragment of pre-tRNAs resulting in tRF-1 formation. The enzyme responsible for the cleavage of tRF-2 fragments is still unclear. The tRF-2 fragment consists of the anticodon loop of the tRNA and has been detected in breast cancer MDA-231 cells [22]. Dicer and ANG cleave tRNAs into ~15–30 nucleotide tRF-3 and tRF-5 fragments. TRF-3 fragments are cleaved at the T loop by Dicer [23] and ANG [24] and tRF-5 fragments are derived from the cleavage of the D-loop by Dicer [23]. The final category of tRFs are i-tRFs, which are internal to the respective tRNA and can straddle the anticodon loop [25]. Limited knowledge of the expression and role of tRNA and tRFs is available in health and disease [26] with even less in musculoskeletal biology [27,28]. 

In this study, we investigated the expression changes of sncRNAs in chondrocytes isolated from healthy metacarpophalangeal joints of young and old horses. Findings for tRNAs are compared to a human OA data set. Age-related changes may predispose cartilage to disease by altering the complex sncRNA expression profile. This provides a sncRNA-wide insight into age-related targets for future therapeutic approaches. 

## 2. Materials and Methods

All reagents were from Thermo-Fisher-Scientific, Loughborough, UK unless stated.

### 2.1. Sample Collection and Preparation

Samples were collected from an abattoir as a by-product of the agricultural industry. Specifically, the Animal (Scientific procedures) Act 1986, Schedule 2, does not define collection from these sources as scientific procedures. Ethical approval was therefore not required. Full thickness equine cartilage was removed from the entire surface of macroscopically normal metacarpophalangeal joints of young *n* = 5 (age mean ± standard deviation; 4 ± 1 years) and old *n* = 5 (17.4 ± 1.9 years) non-Thoroughbred horses. Scoring of the metacarpophalangeal joint was undertaken using a macroscopic grading system, as previously described [29] and samples with no macroscopic perturbations were selected (combined score of zero). Freshly isolated chondrocytes were removed from all 10 samples of harvested cartilage, as previously described [30], plated to confluence and RNA extracted from two million cells per donor. 

In addition to the above samples, RNA from chondrocytes from young *n* = 2 (age mean ± standard deviation; 0.75 ± 0.3 years) and old *n* = 6 (age mean ± standard deviation; 19.3 ± 3.6 years) non-Thoroughbred horses were used for validation.

### 2.2. RNA Isolation, cDNA Library Preparation, and Small RNA Sequencing (RNA-seq)

Total RNA including small RNAs was extracted, as previously described [31], and purified using the miRNAeasy kit (Qiagen, Crawley, UK) according to manufacturer’s instructions and including an on-column DNAse step to remove residual genomic DNA. The integrity of the RNA was assessed on the Agilent 2100 Bioanalyser system using an RNA Pico chip (Agilent, Stockport, UK). The NEBNext® Small RNA Library Prep Set for Illumina® was used for library preparation (New England Biolabs, Ipswich, MA, USA) but with the addition of a Cap-Clip™ Acid Pyrophosphatase (Cell script, Madison, WI, USA) step to remove potential 5′ caps found on some snoRNAs. Samples were amplified for 15 cycles and size was selected. The libraries were sequenced on an Illumina MiSEq platform (Illumina, San Diego, CA, USA) with version 2 chemistry using sequencing by synthesis technology to generate 2 × 150 bp paired-end reads with >12 million clusters per run.

### 2.3. Data Processing

Sequence data were processed through a number of steps to obtain sncRNA expression values including basecalling and de-multiplexing of indexed reads using CASAVA version 1.8.2 [32]; adapter and quality trimming using Cutadapt version 1.2.1 [33] and Sickle version 1.200 to obtain fastq files of trimmed reads; aligning reads to horse genome reference sequences (GCF_002863925.1) using Tophat version 2.0.10 [34] with option “–g 1”; counting aligned reads using HTSeq-count against the annotated features which are combined annotation information from the sources: NCBI *Equus caballus* 3.0 genome annotation, miRBase horse micro RNA annotation, Rfam snoRNA annotation. 

Differential expression analysis was performed in R environment using package DESeq2 [35]. The processes and technical details of the analysis included: assessing data variation and detecting outlier samples through comparing variations of within and between sample groups using principle component analysis (PCA) and correlation analysis; handling library size variation using the DESeq2 default method; formulating data variation using negative binomial distributions; modelling data using a generalised linear model; computing log2 Fold Change (logFC) values for required contrasts based on model fitting results through contrast fitting approach, evaluating the significance of estimated logFC values by Wald test; adjusting the effects of multiple tests using False Discovery Rate (FDR) approach to obtain FDR [36] adjusted P-values. 

### 2.4. Pathway Analysis

In order to identify miRNA targets, bioinformatic analysis was performed by uploading differentially expressed miRNA data into the MicroRNA Target Filter module within Ingenuity Pathway Analysis software (IPA) (Qiagen Redwood City, CA, USA) along with previously identified differentially expressed mRNAs from our ageing equine cartilage study following RNA-seq [31]. In IPA we selected miRNA-target genes based on the direction of differential expression (for example, if a miRNA was reduced in expression it was only matched to mRNAs that demonstrated increased expression). We then identified the networks, functions, and canonical pathways of these miRNA-target genes. 

### 2.5. qRT-PCR Validation

Validation of the small RNA-seq results was undertaken using real-time quantitative PCR (qRT-PCR) analysis in the samples used for sequencing as well as additional samples. SncRNAs were chosen based on level of differential expression. Total RNA was extracted and quantified as above. cDNA was synthesized using 200 ng RNA and the miScript II RT Kit according to the manufacturer’s protocol (Qiagen, Crawley, UK). qRT-PCR mastermix was prepared using the miScript SYBR Green PCR Kit (Qiagen, Crawley, UK) and the appropriate miScript Primer Assay (Qiagen, Crawley, UK) (Appendix A) using 1 ng/μL cDNA according to manufacturer’s guidelines. qRT-PCR was undertaken using a LightCycler® 96 system (Roche, Welwyn Garden City, UK). Relative expression levels were normalised to U6 (as this was stable in the small RNA-seq data set) and calculated using the 2-ΔCt method [37].

### 2.6. tRNA Fragment Analysis

Following the alignment of trimmed reads to NCBI horse genome reference sequences (version 3.0) using Tophat version 2.1.0 [38], the candidate tRNA reads were extracted from the BAM files according to whether they overlapped the ranges covered by tRNA features. The read pairs were stitched into RNA fragments using PEAR (version 0.9.10) [39]. The output reads were aligned to horse tRNA sequences (defined in NCBI GCF_002863925.1_EquCab3.0_genomic.gff) using Bowtie2 version 2.2.5. Only the perfectly mapped fragments were extracted and taken as tRNA fragments for further explorations. Finally, statistical analyses were mainly focused on the fragment length and the mapping start location, which generated the length distribution and the mapping start position distribution of observed tRNA fragments, as well as the summary table for observed tRNA fragments and their target tRNAs. 

### 2.7. Novel snoRNA Analysis

Putative snoRNAs were detected from the raw pair-end reads using ShortStack 3.8.4 [40] with the setting “--mincov 5”, which specifies that the clusters of small RNAs must have at least five alignments. The ShortStack results were subsequently fed into SnoReport 2.0 [41], which uses RNA secondary structure prediction combined with machine learning as the basis to identify the two main classes of snoRNAs; the box H/ACA and the box C/D. Putative snoRNAs were annotated from our experimental small RNA-seq data using ShortStack and SnoReport.

### 2.8. Statistical Analysis

3D PCA score plots and heat maps were carried out using MetaboAnalyst 3.5 [42] (http://www.metaboanalyst.ca) which uses the R package of statistical computing software [43].

For statistical evaluation of qRT-PCR results, following normality testing, Mann-Whitney tests were performed using GraphPad Prism version 7.03 for Windows, (GraphPad Software, La Jolla, CA, USA); p values are indicated.

### 2.9. Human Sample Collection and Preparation

De-identified human OA cartilage approved by Northeast Ohio Medical University (NEOMED) Institutional Review Board and Summa Health Systems, Barberton, Ohio as ‘non-human subject study under 45 CFR’ was used. Total RNA was extracted from smooth, macroscopically intact human OA cartilage with a Mankin score of 2 or less (*n* = 1, female, 60 years old) and damaged OA cartilage with a Mankin score of 4 or higher (*n* = 1, female, 80 years old) using MiRNeasy Kit (Qiagen, Germantown, MD, USA). RNA was quantified using the Nanodrop 1000 Spectrophotometer (Thermo Fisher, Waltham, MA, USA). TapeStation 4200 (Agilent Technologies, Santa Clara, CA, USA) was used to determine RNA integrity using High Sensitivity RNA Screentape analysis kit (Agilent, Santa Clara, CA, USA).

### 2.10. Removal of tRF Modifications from Human Cartilage RNA

The rtStar™ tRF&tiRNA Pretreatment Kit (ArrayStar, Rockville, MD, USA) was used according to the manufacture’s description. For cDNA qRT-PCR library construction, tRF modifications were removed from RNA. The kit removes 3′-aminoacyl and 3′-cP for 3′ adaptor ligation, phosphorylates 5′-OH for 5′-adaptor ligation, and demethylates m1A, m1G, and m3C for efficient cDNA reverse transcription.

### 2.11. tRF 3′ and 5′ Adaptor Ligation for Human Cartilage cDNA Synthesis

The rtStar™ First-Strand cDNA Synthesis Kit (ArrayStar, Rockville, MD USA) was used according to the manufacturer’s description. The kit sequentially ligates 3′-Adaptor with its 5′-end to the 3′-end of the RNAs, and 5′-Adaptor with its 3′-end to the 5′-end of the RNAs. The non-ligation ends of 3′ and 5′ Adaptors were blocked by modifications. A universal priming site for reverse transcription was contained within the 3′ adaptor. Spike-in RNA was used for monitoring the cDNA synthesis efficiency and as a quantitative reference. 

### 2.12. tRF and tRNA Human qPCR Arrays

The nrStar™ Human tRF and tiRNA PCR Array (ArrayStar, Rockville, MD, USA) was used according to the manufacturers description to profile 185 tRF and tiRNA fragments, of which 101 are derived from tRF and tiRNA database [44,45] and the other 84 from recently published papers [46,47,48]. RNA Spike in control and a positive PCR control were used to evaluate PCR efficiency and a genomic DNA control was used to monitor genomic DNA contamination. To profile parent tRNAs, the nrStat *Human* tRNA PCR Array (ArrayStar, Rockville, MD, USA) was used according to the manufacturer’s protocol. This array consisted of 163 PCR-distinguishable nuclear tRNA isodecoders and 22 PCR distinguishable mitochondrial tRNA species covering all anti-codons compiled in GtTNAdb [49,50] and tRNAdb [51] databases. Genomic DNA and positive PCR controls were included to monitor the quality of RNA sample. 

qRT-PCR reactions were conducted using Power SYBR Green master mix (Life technologies, Carlsband, CA, USA) on a Step One Plus (Applied Biosystems, Waltham, MA, USA) machine. U6, SNORD43 and SNORD45 were used as endogenous controls for normalisation of tRF, tiRNA and tRNA detection. Target tRNA, tiRNA and tRF levels were determined as fold change differences utilising the ΔΔCt method [37].

## 3. Results

### 3.1. Preliminary Analysis of Small RNA-seq Data

To identify differential expression of sncRNAs in ageing, Illumina MiSeq was utilised. Summaries of raw, trimmed reads, and mapped reads to *Equus caballus* database can be found in Appendix A. Reads mapping percentages were between 92 to 94.3%. There were 2128 sncRNAs identified. The categories of non-coding RNAs identified are in Figure 1A and Appendix A and included miRNAs, snoRNAs, novel snoRNAs, tRNAs, snRNAs, and long non-coding RNAs (lncRNAs). 

### 3.2. Age-Related Differential Small Non-Coding RNA Gene Expression

There were no overall differences in the distribution of classes of sncRNAs in ageing. The effect of age on the expression of sncRNAs was weak and the separation of young and old samples was not clear. The 3D PCA plot (Figure 1B) indicated that few, but very specific changes in the expression of sncRNAs were found, and samples from old donors were more variable than those from young. A heat map of hierarchical clusters of correlations among samples (Figure 1C) depicts that the sncRNA expression of young and old groups are not very different.

There were 83 sncRNAs differentially expressed with age; six snoRNAs, 11 novel snoRNAs, three snRNAs, 31 lncRNAs, 27 tRNAs (*p* < 0.05) and five miRNAs (FDR-adjusted *p* < 0.05) (Table 1). Data is deposited on NCBI GEO, accession; E-MTAB-8112. 

### 3.3. Age Specific miRNA Interactome

To generate an age-specific miRNA interactome of the most likely miRNA-mRNA target pairs, analysis was performed to identify miRNA targets of the five differentially expressed miRNAs from this study. In IPA, the five miRNAs (miR-143, miR-145, miR-181b, miR-122, miR-148a) were paired with 351 protein coding genes differentially expressed in our previous RNA-seq study on ageing equine cartilage [31]. In Appendix A, the miRNA-mRNA pairings in the correct direction (miR increase and mRNA decrease or vice versa) are shown, including the target predictions and /or experimental validation in the respective database. Four of the five miRNAs interacted in IPA with 31 different differentially expressed target genes reflecting that miRNAs target many mRNAs. These mRNAs targeted by the miRNAs were used in IPA as network-eligible molecules and overlaid onto molecular networks based on information from the IPA Database. Networks for the four miRNAs (miR-148a, miR-122, miR-143, and miR-181b) were generated based on connectivity (Figure 2A–D). Interesting age-related features were determined from the gene networks inferred. Among the top canonical pathways were hepatic fibrosis (*p* = 1.51 × 10^−3^), glycoprotein 6 (GP6) signalling (*p* = 5.67 × 10^−4^), and osteoarthritis pathway (*p* = 2.93 × 10^−3^). The top diseases and cellular functions associated with this network are shown in Table 2 and Figure 3. All IPA results are in Appendix A. 

### 3.4. Confirmation of Differential Gene Expression Using qRT-PCR

For selected snoRNAs (snora46, snora71, snora77, and snord113) and miRNAs (miR-143, miR-145, miR-148a, miR-122, and miR-181b) we used qRT-PCR to validate the small RNA-Seq findings in an extended cohort of chondrocytes. Findings for snora71, snord113, miR-143, miR-145, miR-122, and miR-181b were validated (Figure 4).

### 3.5. tRNA Fragment Changes in Equine Ageing

As we identified differentially expressed tRNAs in ageing equine chondrocytes, we undertook additional analysis to identify tiRNAs and tRFs, as increasing experimental evidence suggests their functional roles in OA [27]. Figure 5 shows cumulative density of tRNA fragment length, alignment length, gene counts, and map start position. On the assessment of data variation using PCA, samples in group “old” did not scatter very closely. Differential expression analysis of all young versus all old did not identify any differentially expressed tRFs (FDR < 0.05). Therefore, we reanalysed the data with the old group divided into two subgroups based on PCA findings: ‘old 1’ (samples 6, 7, 8) and ‘old 2’ (samples 9 and 10) for further differential expression analysis. There were 81 differentially expressed tiRNAs/tRFs; 44 higher in ‘old 2’ and 37 lower in ‘old 2’ compared to young (Appendix A).

### 3.6. Human tRNA and tRF Profiles Compared to Equine tRNA and tRF Profiles

In both human and equine samples, 26 parent tRNAs were detected. Of these 26 tRNAs, 13 tRNAs were induced and 13 were reduced in cartilage from high grade OA compared to cartilage from low grade OA. In equine chondrocytes, 6 parent tRNAs were expressed higher and 20 parent tRNAs were expressed lower in old samples compared to young. Four parent tRNAs were induced in both high grade OA cartilage compared to low grade OA cartilage and in old versus young equine samples. Eleven parent tRNAs were reduced in high grade OA cartilage compared to low grade OA cartilage and in the old versus young equine samples (Figure 6A). 

In both human and equine samples, the tRF-5 fragment known as tRF293/294 and 10 tiRNAs were detected (Figure 6B). In high grade OA compared to low grade OA cartilage, seven tiRNAs and tRF293/294 were induced and three tiRNAs were reduced. In old 1 versus young equine, five tiRNAs and tRF293/294 were induced and five tiRNAs were reduced. In old 2 versus young equine, two tiRNAs were induced and eight tiRNAs and tRF293/294 were reduced. Of these tRNA fragments, three tiRNAs and tRF293/294 were induced and tiRNA-5029-GlyGCC-3 was reduced in high versus low grade OA cartilage and in old 1 versus young equine. Two tiRNAs were induced and three tiRNAs were reduced in high versus low grade OA cartilage and in old 2 versus young equine. In high versus low grade OA cartilage and old 1 versus young equine tiRNA-5029-GlyGCC was reduced.

## 4. Discussion

Our study investigated the changing sncRNA landscape in ageing chondrocytes. Several risk factors exist that influence cartilage health and chondrocyte homeostasis. Among them, ageing is one of the leading risk factors contributing to cartilage-related diseases, such as OA [52]. Many studies have shown that ageing can affect cartilage in different ways, both at cellular and molecular level. Increased chondrocyte death, apoptosis, and a shift towards a catabolic profile have been observed in aged chondrocytes [53]. Additional age-related changes in articular cartilage include increased chondrocyte senescence [54], oxidative stress [55], and changes in the composition and structure of ECM [56]. Although the underlying molecular causes of these changes are not completely understood, it is hypothesised that aged chondrocytes respond differently to various stimuli, such as growth factors, [53,57] and demonstrate altered molecular signatures [6]. 

SncRNAs are a subset of epigenetic modifiers and their role in cartilage ageing has been studied increasingly in the last decade [6,31,58]. In the current study, we have used small RNA-seq to identify alterations in sncRNAs between young and old equine chondrocytes. The horse is a good model to study musculoskeletal ageing and disease as we could assess the whole joint for pathological perturbations during tissue collection. It is very challenging to source aged human cartilage that has no OA changes, whereas this is easily undertaken in equine samples. Moreover, the horse has been used as a model of OA and there has been significant research on equine joint anatomy and pathophysiology [59,60].

Within our set of differentially expressed sncRNAs, miRNAs are the best studied in musculoskeletal ageing and cartilage. In old chondrocytes, we identified two miRNAs with higher expression; miR-122 and miR-148a, and three miRNAs with lower expression; miR-143, miR-145, and miR-181b. Of these five miRNAs, all except miR-148a were validated in an extended cohort of young and old equine chondrocytes with qRT-PCR. MiR-122 has been researched extensively in liver [61,62], but its role in musculoskeletal ageing is less clear. MiR-122 was decreased in the serum and plasma of patients with osteoporosis, the most common age-related bone disease [63], but was significantly upregulated in senescent human fibroblasts [64] and was shown to upregulate p53 which is induced in senescence [65]. MiR-143 was downregulated in muscle satellite cells from old mice and primary myoblasts from old humans and mice [66]. In addition, circulating miR-143 was upregulated in young individuals following resistant exercise, but was downregulated in older individuals after resistant exercise [67]. MiR-145 was downregulated in old OA patients [68] as well as in experimental OA rat chondrocytes treated with tumour necrosis factor (TNF) [69] and, finally, miR-181b was downregulated in skeletal muscle of old rhesus monkeys [70]. 

To further investigate the potential role of the differentially expressed miRNAs identified in this study, we used IPA to combine them with the differentially expressed mRNAs from our previous equine cartilage study [31]. IPA miRNA ‘Target Filter and Expression Pairing’ identified 31 potential target genes. IPA core analysis of these genes revealed canonical pathways associated with cartilage physiology, such as role of chondrocytes in rheumatoid arthritis, OA-related pathways and bone morphogenic protein (BMP) signalling, all of which have been reported to change with ageing [56,71,72]. Moreover, top diseases and disorders linked to these genes, as identified by IPA, included skeletal and muscular disorders and connective tissue disorders. Of note, follistatin (FST) which was upregulated in old equine cartilage and was predicted by IPA as a target of the downregulated miR-143, was overexpressed in human OA chondrocytes [73] and canine OA cartilage [74] and was induced by telomere shortening [75], a process associated with ageing. Moreover, *TNF ligand superfamily member 11 (TNFSF11)*, also known as *receptor activator of nuclear factor kappa-Β ligand (RANKL)* was upregulated in old equine cartilage and was identified as a predicted target of the downregulated miR-181b. Higher expression of *TNFSF11/RANKL*, which correlated with bone loss, was reported in old C57BL/6 mice [76], rabbits with chronic antigen-induced arthritis [77], and in human high grade OA cartilage [78]. These results demonstrate the adverse effect of ageing on miRNA levels and their potential use as biomarkers or therapeutic targets for age-related musculoskeletal diseases. 

Six snoRNAs were identified as differentially expressed due to ageing in chondrocytes. This conserved class of non-coding RNAs are principally characterised as guiding site-specific post-transcriptional modifications in ribosomal RNA [79] (canonical snoRNAs), but can also modify additional classes of RNAs including other snoRNAs, tRNAs and mRNAs; so called non-canonical snoRNAs [80,81]. Examples of age-related snoRNAs in equine ageing chondrocytes with canonical functions include snora71 and snord29. Novel non-canonical functions reported for snoRNAs including the modulation of alternative splicing [82], an essential involvement in stress response pathways [83] and the modulation of mRNA 3′ end processing [84]. Similar to miRNAs, snoRNAs are emerging as important regulators of cellular function and disease development [15], related to their ability to fine-tune ribosomes accommodating changing requirements for protein production during development, normal function, and disease [85]. We have previously identified a role for snoRNAs in cartilage ageing and OA [16] and their potential use as biomarkers for OA [9]. Interestingly there was a reduction in snord113/114 in ageing chondrocytes which agrees with our previous findings in equine cartilage ageing [31]. We have also previously demonstrated that SNORD113 was reduced in ageing human knee cartilage but increased in OA [16] and increased in human anterior cruciate ligament [17]. SNORD113/114 are located in imprinted loci and may play a role in the evolution and/or mechanism of epigenetic imprinting. Although belonging to the C/D box class of snoRNAs which direct site-specific 2′-O-ribose methylation of substrate RNAs, they differ from other C/D box snoRNAs in their tissue specific expression profiles (including fibroblast, chondrocytes and osteoblasts) and the lack of known substrate RNA complementarity. This currently classifies them as orphan snoRNAs as they are not predicted to guide the 2′-O-ribose methylation but have novel, unknown roles [86]. Additionally, SNORD113 functions as a tumour suppressor in hepatic cell carcinoma by reducing cell growth, and it inactivates the phosphorylation of extracellular signal-regulated kinase (ERK) 1/2 and mothers against decapentaplegic homolog (SMAD) 2/3 in mitogen-activated protein kinase (MAPK)/ERK and transforming growth factor beta (TGF-β) pathways [87]. Together, our snoRNA findings indicate that age-related changes in chondrocyte snoRNAs could have important implications through both canonical and non-canonical snoRNA routes.

This is the first study to detect tRNA and tRNA fragments in equine chondrocytes and to compare these findings with tRNA and tRNA fragments detected in human OA cartilage. For our tRNA data, old donors clustered into two groups and further analysis was undertaken with these subgroups. There were no apparent differences in these subgroups with regards to age or sex and scores were all zero. The parent tRNA Cys-GCA was found to be increased in both aged equine chondrocytes and high grade OA human cartilage samples. tRNA Cys-GCA levels have previously been reported to be increased in human chondrocytes induced with the cytokine interleukin (IL) 1 beta resulting in the production of the tRNA fragment tRF-3003a, a type 3-tRF produced by the cleavage of Cys-GCA. tRF-3003a has been shown to post-transcriptionally regulate the Janus Kinase 3 (JAK3) expression through sequence complementarity via the Argonaute (AGO) / RNA-induced silencing complex (RISC) in human chondrocytes [27].

The Janus Kinase and Signal Transducer and Activator of Transcription (JAK-STAT) pathway is the target of several cytokines such as interferon-γ, IL-2, IL-4, IL-6, IL-7, IL-10, IL-12, and IL-15. Many of these cytokines are known to play important roles in synovial inflammation during OA pathogenesis [88,89].

The tRNA fragments detected in equine samples that matched with human samples consisted of 5′ tiRNA and tRF-5 fragments. Many of the equine fragments detected did not fall into the classical tRF-3, tRF-5, or tiRNA size ranges and instead may likely be i-tRFs, which are internal to the respective tRNA and can straddle the anticodon loop. 5′ tiRNA, 3′ tiRNA, tRF-3, and tRF-5 fragments were detected in the low-grade OA cartilage and in the high grade OA cartilage. In our human studies, base modifications found on tRNAs and tRF/tiRNA fragments that would normally block reverse transcription were removed and this may account for some of the differences found between the equine and human tRNA/tRF profiles.

The importance of modulation of tRNA levels and tRNA fragments in articular cartilage homeostasis remains an unexplored area. This is the first evidence that aged equine samples have changes in the expression of specific tRNAs and tRFs when compared to young equine samples. We report for the first time several 5′ tiRNAs, such as tiRNA Glu-TTC and tiRNA His-GTG, were induced in old compared to young equine chondrocytes and in high grade compared to low grade human OA cartilage. Previous reports have shown that 5′ tiRNAs can be produced by cell stress in mammalian cells and these 5′ tiRNA half fragments may have a role in inhibiting cell translation and could be involved in stress granule formation [90]. Further studies are required to find the mechanism by which these fragments are produced and whether the changes in the profile of fragments found in old compared to young equine chondrocytes or high compared to low grade OA cartilage potentially contribute to the development of OA.

We are aware our study has a number of limitations. The effect of ageing between young and old equine chondrocytes was small on the differential expression of sncRNAs. However, it is likely that we are therefore interrogating highly specific changes that are age dependent. Furthermore, we cannot rule out changes related to the use of chondrocytes instead of cartilage tissue. Even though chondrocytes of low passage were used, collagenase digestion and plating of cells could have affected their phenotype and gene expression. The choice of chondrocytes over cartilage tissue was made based on RNA from cartilage tissue is of low quality (in our hands) and heavily contaminated with proteoglycans [91], usually making it challenging for sequencing. In our previous study interrogating snoRNAs in human knee cartilage ageing and OA [16] we utilized cartilage tissue as opposed to chondrocytes. The old non-OA cartilage was derived from the grossly normal condyle of OA joints. In the study we found a number of age and OA specific snoRNAs, but we could not define the cartilage as truly normal as it was derived from an OA joint. Nevertheless, we identified two age related, no species specific snoRNAs that were DE in both our studies, snord113 and snord29. The use of the Illumina MiSeq platform would have contributed to the low number of differentially expressed sncRNAs as it offers less depth coverage compared to other platforms, such as the HiSeq platform. Finally, we have used a relatively small number of samples per group. Given the use of primary cells and the degree of variability observed, especially for the old group, the inclusion of five samples pre group may have contributed to the small number of differentially expressed sncRNAs in ageing equine chondrocytes on our study.

## 5. Conclusions

For the first time we have described, using unbiased methods, the effect of ageing on the expression of sncRNAs in equine chondrocytes. We detected variable classes of sncRNAs (the small non-coding RNAome) in young and old chondrocytes, which are differentially abundant, indicating that there are multiple levels of epigenetic control in cartilage and chondrocyte ageing. Among them, there were miRNAs, which are predicted to play a role in the development of the musculoskeletal system and in skeletal disorders. In addition, the current study is one of the few studies that have investigated tRNAs and tRNA fragments, in an attempt to uncover novel molecular signatures in aged and diseased chondrocytes/cartilage that could be useful in the future as therapeutic targets. Further research is needed to elucidate the role and function of these molecules and their potential link to disease. 

## Figures and Tables

**Figure 1 ijms-21-05675-f001:**
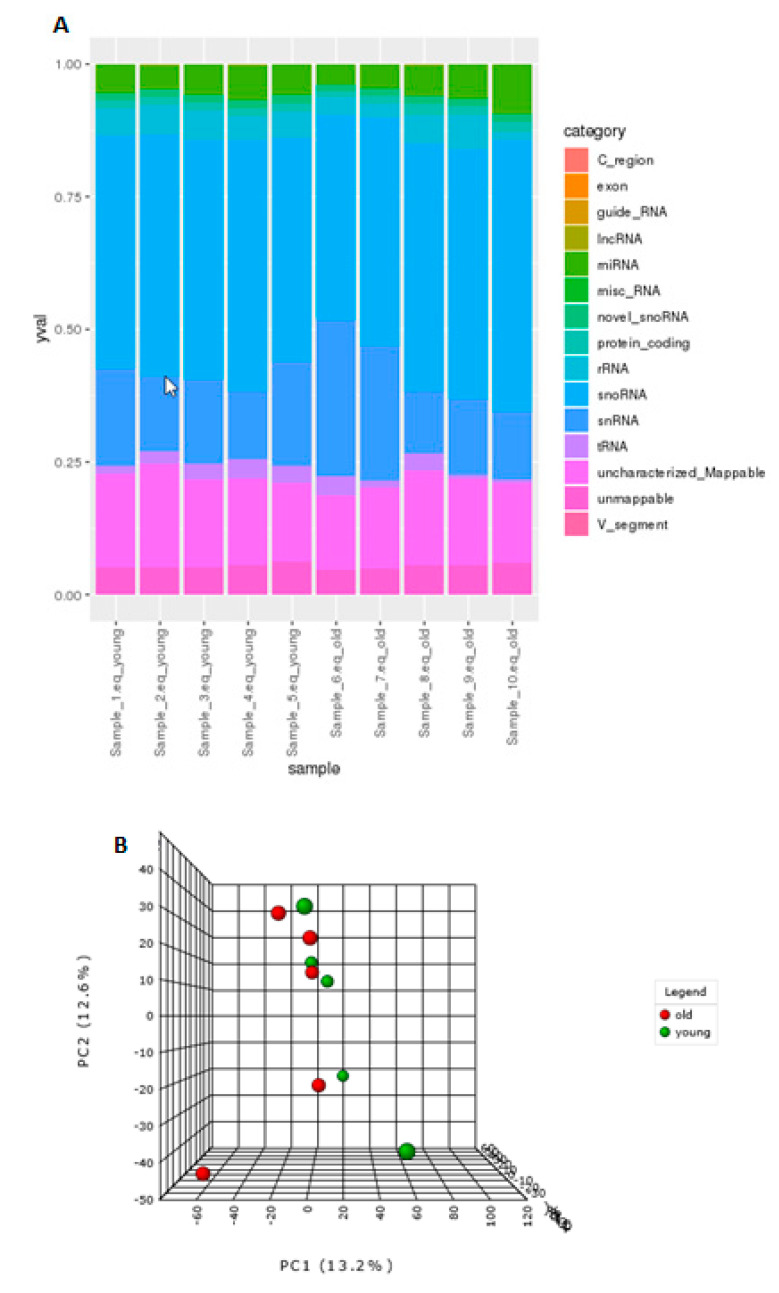
Age-related differential small non-coding RNA (sncRNA) gene expression. (**A**) The categories of non-coding RNAs identified in young and old equine chondrocytes; long non-coding RNAs (lncRNAs), microRNAs (miRNAs or miRs), small nucleolar RNAs (snoRNAs), small nuclear RNAs (snRNAs), transfer RNAs (tRNAs). (**B**) 3D principle component analysis (PCA) plot between the selected principle components (PCs). The explained variances are shown in brackets. (**C**). Clustering results shown as a heatmap (distance measure using Euclidean, and clustering algorithm using Ward) for the top 90 molecules.

**Figure 2 ijms-21-05675-f002:**
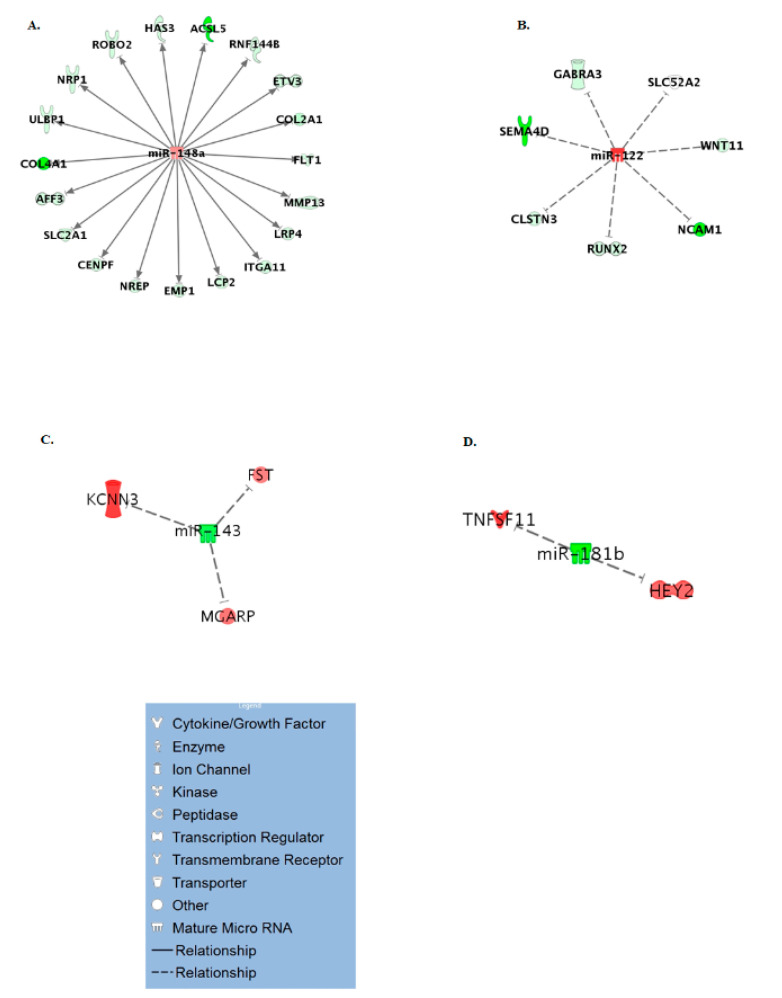
miR-mRNA interactome for differentially expressed miRs in ageing. Significantly differentially expressed miRs were paired with differentially expressed mRNAs from our original cartilage study. (**A**) miR-148a, (**B**) miR-122, (**C**) miR-143, (**D**) miR-181b. The legend for individual molecules is shown. Genes in red are upregulated and green downregulated in old chondrocytes/cartilage compared to young, and depth of colour correlates to fold expression change.

**Figure 3 ijms-21-05675-f003:**
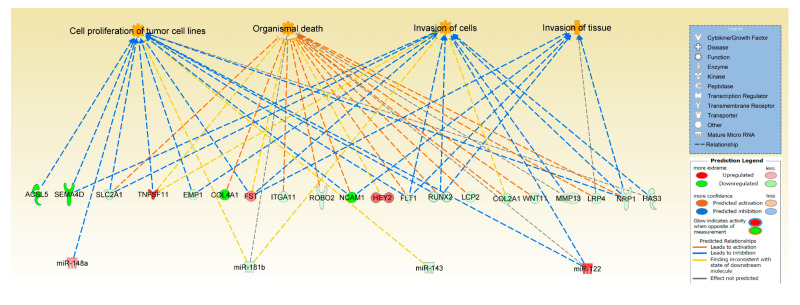
Significant pathways and networks affected in cartilage ageing. IPA was used to pair differentially expressed mRNA and miRNA data from ageing equine cartilage and chondrocytes. Figure is a graphical representation between molecules identified in our data in their respective networks. Red nodes; upregulated gene expression in old group; green nodes; downregulated gene expression in old group. Intensity of colour is related to higher fold-change. The key to the main features in the networks is shown.

**Figure 4 ijms-21-05675-f004:**
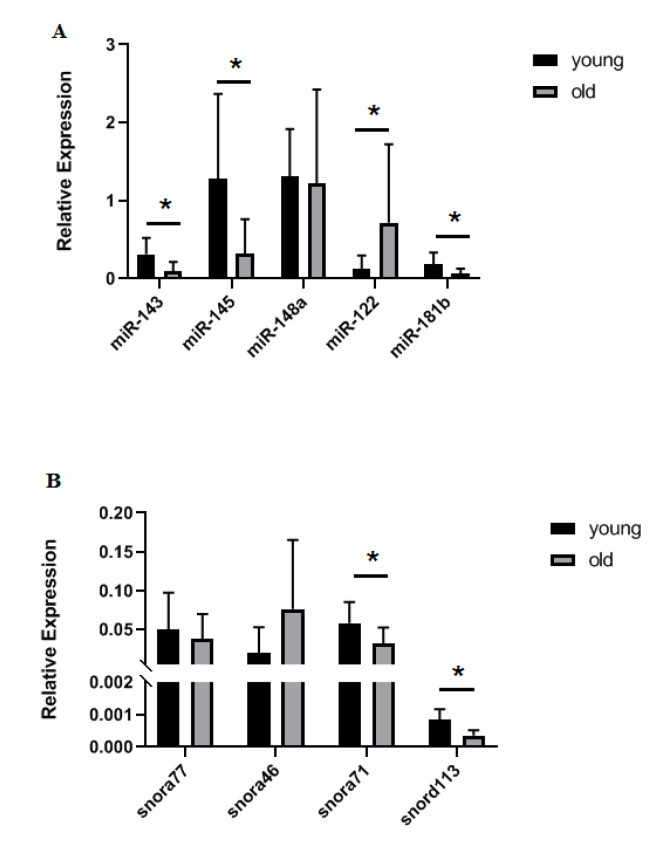
Validation of selected sncRNAs in an extended cohort of equine young and old chondrocytes. Real-time quantitative Polymerase Chain Reaction (qRT-PCR) was used to validate findings from the small RNA sequencing (**A**) microRNAs and (**B**) snoRNAs, *n* = 5–12 per group. Statistical analysis was undertaken using a Mann Whitney test in GraphPad Prism. Mean and standard errors are shown with * denoting *p* < 0.05.

**Figure 5 ijms-21-05675-f005:**
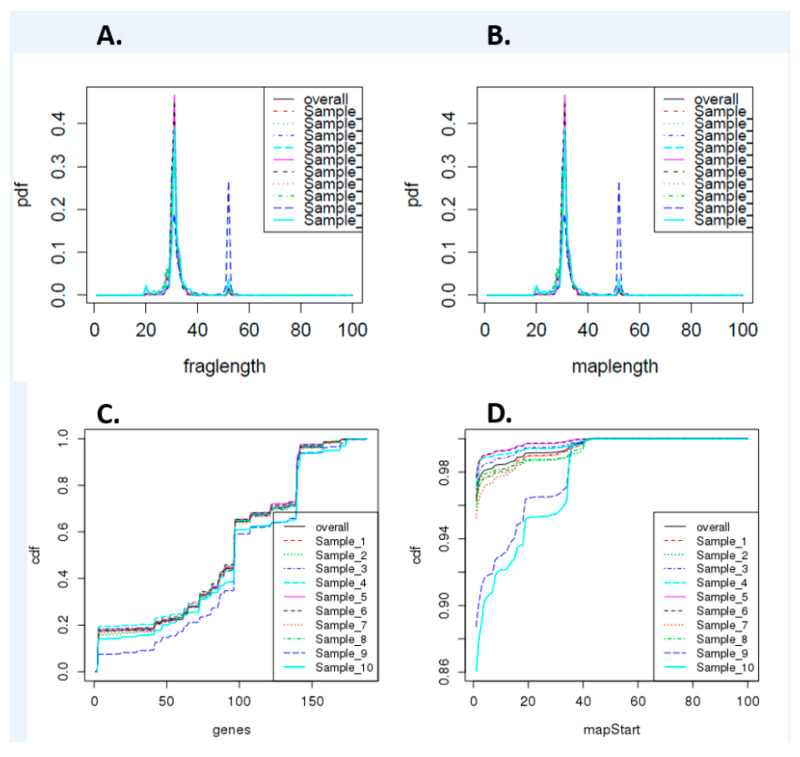
Summary of differentially expressed tRNA fragment data. (**A**) Cumulative density of tRNA fragment length, (**B**) alignment length, (**C**) gene counts, and (**D**) map start position. Samples 1–5 are derived from young donors and 6–10 are old donors.

**Figure 6 ijms-21-05675-f006:**
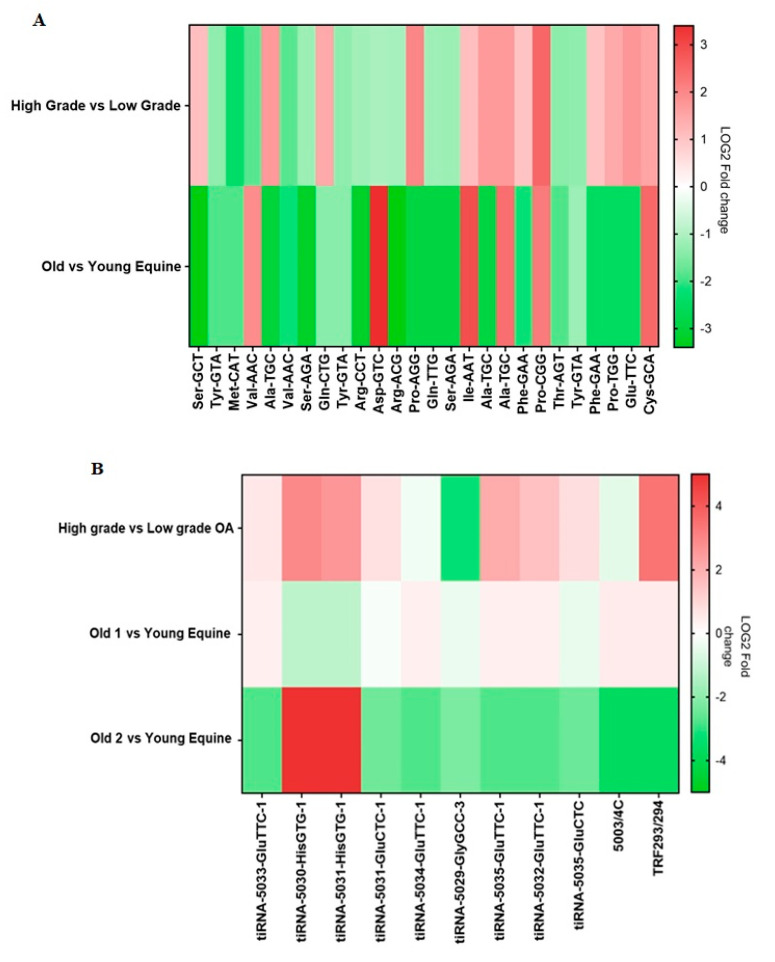
tRNA analysis of human osteoarthritic cartilage. (**A**) High grade versus low grade OA human cartilage and old versus young equine chondrocyte tRNA profiles. Heatmap of Log2 fold change expression of 26 tRNAs detected in both human and equine samples. Human cartilage tRNAs detected using a Human tRNA PCR Array. Red highlights induced tRNA expression in high grade OA human cartilage compared to low grade and in old equine chondrocytes compared to young. Green highlights reduced tRNA expression in high grade OA human cartilage compared to low grade and in old equine chondrocytes compared to young. (**B**) High grade OA versus low grade OA human cartilage and equine tiRNA/tRF profiles. Heatmap of Log2 foldchange expression of 11 tRF/tiRNA fragments detected in both human and equine samples. Human cartilage tRF/tiRNA detected using a Human tRF&tiRNA PCR Array. Red highlights induced tRF/tiRNA expression in human high grade versus low grade OA samples and in old versus young equine samples. Green highlights reduced tRF/tiRNA expression in human high grade versus low grade OA samples and in old versus young equine samples.

**Table 1 ijms-21-05675-t001:** Differentially expressed sncRNAs in ageing chondrocytes. Log_2_ fold change values were derived with young as the reference group. A positive log_2_ fold change equates to higher expression in old, whereas a negative log_2_ fold change equates to lower expression in old. All are significant at *p* < 0.05, except for miRNAs that are significant with a false-discovery rate (FDR-)adjusted p value of *p* < 0.05.

Gene/Transcript Name	Gene Biotype	Log_2_ Fold Change
eca-miR-143	miRNA	−1.3
eca-miR-145	miRNA	−1.8
eca-miR-181b	miRNA	−1.8
eca-miR-122	miRNA	2.3
eca-miR-148a	miRNA	1.3
snora71	snoRNA	−3.2
snord113	snoRNA	−2.4
snora46	snoRNA	1.0
snora77	snoRNA	2.0
snora47	snoRNA	2.5
snord29	snoRNA	2.5
ECABCGRLG0000003960	novel snoRNA	−2.6
ECABCGRLG0000002980	novel snoRNA	−2.5
ECABCGRLG0000006050	novel snoRNA	−1.6
ECABCGRLG0000000640	novel snoRNA	−1.1
ECABCGRLG0000002800	novel snoRNA	−1.0
ECABCGRLG0000007680	novel snoRNA	1.8
ECABCGRLG0000008070	novel snoRNA	2.0
ECABCGRLG0000007010	novel snoRNA	2.5
ECABCGRLG0000008090	novel snoRNA	2.9
ECABCGRLG0000004470	novel snoRNA	3.0
ECABCGRLG0000005680	novel snoRNA	3.1
LOC111775808	snRNA	−1.3
LOC111773055	snRNA	2.2
LOC111772636	snRNA	2.5
LOC111770368	lncRNA	−3.4
LOC111768432	lncRNA	−3.2
LOC102148414	lncRNA	−3.2
LOC102149168	lncRNA	−2.9
LOC111772155	lncRNA	−2.5
LOC106783307	lncRNA	−2.5
LOC111775319	lncRNA	−2.5
LOC102148711	lncRNA	−2.5
LOC111775759	lncRNA	−2.5
LOC111774351	lncRNA	−2.5
LOC102150704	lncRNA	−2.5
LOC111775994	lncRNA	−2.5
LOC102150027	lncRNA	−2.5
LOC106781358	lncRNA	−2.5
LOC102147393	lncRNA	−2.5
LOC111776203	lncRNA	−2.5
LOC111771286	lncRNA	−1.3
LOC102149893	lncRNA	1.7
LOC111775969	lncRNA	1.7
LOC106781629	lncRNA	1.9
LOC111773181	lncRNA	2.0
LOC102149863	lncRNA	2.5
LOC102149361	lncRNA	2.5
LOC111770630	lncRNA	2.5
LOC102147707	lncRNA	2.5
LOC106783385	lncRNA	2.5
LOC102149569	lncRNA	2.5
LOC106782740	lncRNA	2.5
LOC111770896	lncRNA	2.5
LOC102150024	lncRNA	2.9
LOC102150338	lncRNA	2.9
TRNAR-ACG	tRNA	−3.4
TRNAR-CCU	tRNA	−3.2
TRNAS-AGA	tRNA	−3.2
TRNAS-GCU	tRNA	−3.1
TRNAA-UGC	tRNA	−3.0
TRNAP-AGG	tRNA	−2.9
TRNAQ-UUG	tRNA	−2.9
TRNAS-AGA	tRNA	−2.9
TRNAA-UGC	tRNA	−2.9
TRNAF-GAA	tRNA	−2.5
TRNAP-UGG	tRNA	−2.5
TRNAE-UUC	tRNA	−2.5
TRNAF-GAA	tRNA	−2.2
TRNAV-AAC	tRNA	−2.2
TRNAM-CAU	tRNA	−2.1
TRNAT-AGU	tRNA	−1.9
TRNAY-GUA	tRNA	−1.9
TRNAQ-CUG	tRNA	−1.4
TRNAN-GUU	tRNA	−1.4
TRNAY-GUA	tRNA	−1.4
TRNAY-GUA	tRNA	−1.2
TRNAV-AAC	tRNA	1.9
TRNAP-CGG	tRNA	2.2
TRNAA-UGC	tRNA	2.4
TRNAC-GCA	tRNA	2.5
TRNAI-AAU	tRNA	2.9
TRNAD-GUC	tRNA	3.4

**Table 2 ijms-21-05675-t002:** IPA mRNA target diseases and functions.

a. Top Molecular and Cellular Functions	*p*-Value Range	Number of Molecules
Cellular Movement	1.41 × 10^−3^–4.23 × 10^−10^	21
Cell Morphology	1.37 × 10^−3^–3.52 × 10^−8^	16
Cellular Development	1.42 × 10^−3^–4.94 × 10^−7^	24
Cell Death and Survival	1.37 × 10^−3^–1.08 × 10^−5^	18
Cell-To-Cell Signalling and Interaction	1.37 × 10^−3^–1.88 × 10^−5^	20
**b. Top Diseases and Disorders**		
Skeletal and Muscular Disorders	1.50 × 10^−3^–1.22 × 10^-9^	16
Connective Tissue Disorders	1.50 × 10^−3^–3.52 × 10^-8^	16
Organismal Injury and Abnormalities	1.50 × 10^−3^–3.52 × 10^-8^	31
Cancer	1.49 × 10^−3^–5.30 × 10^-7^	31
Developmental Disorder	1.37 × 10^−3^–5.30 × 10^-7^	17

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
