# Peer review of "Small Non-Coding RNAome of Ageing Chondrocytes"

_ijms, 2020, doi:10.3390/ijms21165675_

Round 1

Reviewer 1 Report

This is an extremely interesting study about ageing effect on the expression of small Non-coding RNAome of chondrocytes. This manuscript is very well written. In my opinion, the current manuscript meets the quality guidelines to be published in Int. J. Mol. Sci., just with a few minor comments detailed below.

  1. As the authors mentioned in the last paragraph, one of the limitations of this study is the use of chondrocytes instead of cartilage tissue. It could be interesting if the authors compare and add the discussion related to ‘SnoRNA signatures in cartilage ageing and osteoarthritis’, where cartilage was using.
  2. The Authors should pay attention to the format and typos of ms. I list just a few below as examples. Ex:The space after ‘Liu’ should be deleted (line 4), 'loadbearing' should be load-bearing or load bearing’ (line 37), what does the tissue h mean? (line 38).

Author Response

As the authors mentioned in the last paragraph, one of the limitations of this study is the use of chondrocytes instead of cartilage tissue. It could be interesting if the authors compare and add the discussion related to ‘SnoRNA signatures in cartilage ageing and osteoarthritis’, where cartilage was using.

Author’s response

We have already discussed some of the cross over findings between results in our recently published study in Scientific Reports (SnoRNA signatures in cartilage ageing and osteoarthritis) from line 426. However, we have now also further in the discussion in the final paragraph as kindly suggested by the reviewer.

The Authors should pay attention to the format and typos of ms. I list just a few below as examples. Ex:The space after ‘Liu’ should be deleted (line 4), 'loadbearing' should be load-bearing or load bearing’ (line 37), what does the tissue h mean? (line 38).

Author’s response

We thank the authors for noticing these typos in the initial version. We have been through the manuscript and corrected all the errors pointed out by the reviewer and additional ones we found.

Reviewer 2 Report

Congratulations for the authors for such an interesting and innovative work. I just want to point out a few details:

  1. in the methods, part 2.1: "freshly isolated chondrocytes were isolated from harvested cartillage" ..., but does not describe if all (five young and five old) samples were used for isolating of chondrocytes. Even though you point out that additional samples were used for validation.
  2. In the methods about RNA isolation "the miRNAeasy kit" noted, I assume the same kit was used for the isolation of small RNA fragments.
  3. Figure 1A is indicated on page 6, but figure A, B or C is not indicated in figure 1.
  4. I suggest looking for the expression on the page 6 (a heat map .... are not very different), in the next sentence you point out all the differences that have been found.
  5. You point to Tables 1 and 2, but do not have them indicated. As in Figure 4, you point out that n = 5-12 per group, then you validate selected sncRNAs in an extended cohort of two groups of chondrocytes. It would be good to clarify where these groups come from.
  6. On page 10 you point out that to find the differences between the young and old groups it was necessary to subdivide the group of the old ones. It could indicate if there is any difference between these subgroups (sex, age, scores, etc.).
  7. In figure 7 it is necessary to indicate the parts A and B.

Finally, just to point out that there are some details in the writing (what means “tissue h of ...) on page 1, line 38), in some lines they leave more than one space between the words: lines 43, 263, 452, for example.

Author Response

In the methods, part 2.1: "freshly isolated chondrocytes were isolated from harvested cartillage" ..., but does not describe if all (five young and five old) samples were used for isolating of chondrocytes. Even though you point out that additional samples were used for validation.

Author’s response

We apologise if this was not clear and have amended the methods to make it clear that all ten samples were used (line 120).

In the methods about RNA isolation "the miRNAeasy kit" noted, I assume the same kit was used for the isolation of small RNA fragments.

Author’s response

We thank the reviewers for pointing out that this statement was unclear and have amend the sentence (line 126) accordingly as we did indeed use the kit to isolate small RNA fragments.

Figure 1A is indicated on page 6, but figure A, B or C is not indicated in figure 1.

Author’s response

We apologise that we did not insert A, B, C onto the figures 1A, 1B, 1C. We have now amended this.

I suggest looking for the expression on the page 6 (a heat map .... are not very different), in the next sentence you point out all the differences that have been found.

Author’s response

We thank the reviewer for this comment. Indeed there are few differences on a global scale in sncRNAs with only groups than 83 age-related sncRNAs in total different. Therefore, based on the heatmap we say in the manuscript that these samples are not very different. However, in the next sentence we mention the differences that we actually did find. Also, except for miRNAs, differentially expressed sncRNAs were identified using p value<0.05 instead of the stricter p-adjusted FDR <0.05, which justifies our claim about the two groups not being very different in their sncRNA expression. We hope this explanation is sufficient.

You point to Tables 1 and 2, but do not have them indicated. As in Figure 4, you point out that n = 5-12 per group, then you validate selected sncRNAs in an extended cohort of two groups of chondrocytes. It would be good to clarify where these groups come from.

Author’s response

We apologise, as tables 1 and 2 do not appear to be in the original manuscript but are now included.

The extended cohort of chondrocytes consists of the same samples that were used for small RNAseq (5 young and 5 old) as well as new donors, as mentioned in methods section 2.1. This was undertaken to increase the number of donors for qRT-PCR validation. However, due to limited RNA quantity from some donors, we did not test all miRs and snoRNAs shown in figure 4 for all the donors of the extended cohort. This is why the n number fluctuates from 5 to 12.

On page 10 you point out that to find the differences between the young and old groups it was necessary to subdivide the group of the old ones. It could indicate if there is any difference between these subgroups (sex, age, scores, etc.).

Author’s response

We did indeed look to see sex, or age (all scores were zero so this was not a factor) caused the clustering into two groups. We did not find any reason from the data that we had. We have added a sentence re this in the discussion (line 445).

In figure 7 it is necessary to indicate the parts A and B.

Author’s response

I believe the reviewer is referring to figure 6 as there is not figure 7. We have added the letters on each figure.

Finally, just to point out that there are some details in the writing (what means “tissue h of ...) on page 1, line 38), in some lines they leave more than one space between the words: lines 43, 263, 452, for example.

Author’s response

We apologize for these errors and they have all been amended.